# Electronic structure of the candidate 2D Dirac semimetal SrMnSb$_2$: a combined experimental and theoretical study

Shyama V. Ramankutty[1*], Jans Henke[1,2], Adriaan Schiphorst[1], Rajah Nutakki[1], Stephan Bron[1], Georgios Araizi-Kanoutas[1], Shrawan K. Mishra[3], Lei Li[4], Yingkai Huang[1], Timur K. Kim[5], Moritz Hoesch[5], Christoph Schlueter[5], Tien-Lin Lee[5], Anne de Visser[1], Zhicheng Zhong[4], Jasper van Wezel[2], Erik van Heumen[1] and Mark S. Golden[1‡]

**1** Van der Waals Zeeman Institute, IoP, University of Amsterdam, The Netherlands
**2** Institute for Theoretical Physics, IoP, University of Amsterdam, The Netherlands
**3** School of Material Science and Technology, Indian Institute of Technology (BHU), Varanasi 221005, India
**4** Key Laboratory of Magnetic Materials and Devices, Ningbo Institute of Materials Technology and Engineering, Chinese Academy of Sciences
**5** Diamond Light Source Ltd., Harwell Science and Innovation Campus, Didcot, UK

⋆ S.V.Ramankutty@uva.nl,       ‡ M.S.Golden@uva.nl

## Abstract

SrMnSb$_2$ is suggested to be a magnetic topological semimetal. It contains square, 2D Sb planes with non-symmorphic crystal symmetries that could protect band crossings, offering the possibility of a quasi-2D, robust Dirac semi-metal in the form of a stable, bulk (3D) crystal. Here, we report a combined and comprehensive experimental and theoretical investigation of the electronic structure of SrMnSb$_2$, including the first ARPES data on this compound. SrMnSb$_2$ possesses a small Fermi surface originating from highly 2D, sharp and linearly dispersing bands (the 'Y-states') around the $(0,\pi/a)$-point in $k$-space. The ARPES Fermi surface agrees perfectly with that from bulk-sensitive Shubnikov de Haas data from the same crystals, proving the Y-states to be responsible for electrical conductivity in SrMnSb$_2$. DFT and tight binding (TB) methods are used to model the electronic states, and both show good agreement with the ARPES data. Despite the great promise of the latter, both theory approaches show the Y-states to be gapped *above* E$_F$, suggesting trivial topology. Subsequent analysis within both theory approaches shows the Berry phase to be zero, indicating the non-topological character of the transport in SrMnSb$_2$, a conclusion backed up by the analysis of the quantum oscillation data from our crystals.

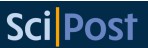

## 1 Introduction

An important group within the class of topologically non-trivial systems of fermions are the Dirac semi-metals. Inspired by theoretical predictions [1–3], experimental searches have been carried out for 3D Dirac semimetals (3D-DSM). ARPES data have shown $Na_3Bi$ [4] and $Cd_3As_2$ [5,6] to be 3D-DSM. Addition of magnetism (breaking time reversal symmetry [TRS]) or violation of inversion symmetry can introduce Weyl physics in such systems [7, 8]. The latter has been shown recently in systems like TaAs [9–11] and NbAs [12]. Only very recently has a TRS breaking, magnetic Weyl system been proposed in the canted anti-ferromagnet $YbMnBi_2$ based upon ARPES experiments and electronic structure calculations [13,14]. These developments indicate the intrinsic richness of condensed matter systems in delivering novel quasiparticles with no known or verified counterparts elsewhere in physics.

 Recently $SrMnSb_2$, with a crystal structure comparable to that of $YbMnBi_2$, has been proposed [15] to be a magnetic topological semimetal from the combined results of Shubnikov de Haas (SdH) and de Haas van Alphen (dHvA) experiments, as well as magnetometry and neutron scattering. At low temperature, $SrMnSb_2$ is a canted anti-ferromagnet, and the same is true for $YbMnBi_2$ [13]. Comparing to DFT calculations on $SrMnSb_2$ [16] and based on the similarities with $YbMnBi_2$, the proposal has been made that there are linearly dispersing electronic states at the Y-point in the Brillouin zone [15], and that these states are topologically non-trivial, supported by a Landau level analysis of the SdH oscillations, leading to the extraction of a $\pi$-Berry phase. In [15], the band gap seen at $E_F$ for the states at the Y-point in the DFT data [16] is argued to close in real crystals, due to the fact that the true $SrMnSb_2$ crystal structure is slightly different to that used in the published DFT [16], and, as stated in [15] due to breaking TRS. Interestingly, both $YbMnBi_2$ and $SrMnSb_2$ possess a non-symmorphic crystal structure. Non-symmorphic symmetries are an added reason for interest in this class of materials, as they have been shown theoretically to protect 2D Dirac crossings [17]. Although the experimental search for such materials has been the focus of much recent work, there are only a handful of materials that have been shown experimentally to host such protected 2D Dirac fermions, such as members of the ZrSiX (X=S, Se and Te) [18–20] family.

The aim of this paper is threefold: to experimentally probe the electronic structure of $SrMnSb_2$ using ARPES for the first time, combining this information with magnetotransport data from the same crystals, so as to decide whether Y-states are responsible for the novel transport data. If so, deciding whether these states are topologically non-trivial forms the second aim of this paper, and we combine experiment and theory input on this point. Finally, we combine first principles and tight binding theoretical approaches to explore the possible existence of non-symmorphic protected states in $SrMnSb_2$.

The paper is organised as follows. In the next section, the experimental and theoretical methods used are described. The results and discussion part of the paper deals first with the crystal structure, then the transport, magnetic and surface characterisation of $SrMnSb_2$. ARPES data is presented and discussed next, followed by the density functional theory and tight binding analyses and their comparison with the ARPES data. Then, an analysis of the Berry phase for the electronic states of $SrMnSb_2$ relevant for transport is presented, together with a discussion of the role of magnetism in this compound. The paper closes by bringing together the conclusions of the study as a whole and includes two appendices. The first describes the details of the tight binding calculations, and the second discusses additional ARPES experiments carried out in order to shift up the chemical potential of $SrMnSb_2$ by *in situ* adsorption of potassium at low temperature.

## 2 Methods

### I. Experimental details

High quality single crystals of $SrMnSb_2$ were grown at the University of Amsterdam, using a self-flux growth method starting from a stoichiometric mixture of Sr, Mn and Sb, as reported in [15]. Transport experiments were carried out in a Quantum Design PPMS, as were measurements of the magnetic susceptibility, the latter in a field of 2 T, applied in the in-plane direction (for a schematic of the crystal structure, see Fig. 1, below). Shubnikov-de Haas oscillation measurements were also carried out using the PPMS at a temperature of 2 K and in magnetic fields up to ±9 T with the field applied in the out-of-plane direction. As the surface sensitivity of ARPES requires cleavage of the crystal prior to measurement, we conducted low energy electron diffraction (LEED) experiments, using a reverse-view LEED system mounted in the preparation chamber of the lab-based spectrometer in Amsterdam.

The ARPES data presented here were measured at the I05 [I09] beamline in Diamond Light Source Ltd. (Didcot, U.K.), and at the $1^3$-end station at the BESSY-II synchrotron facility, part of the HZB in Berlin. At I05 [I09], the vacuum was kept below $5 \times 10^{-11} [2 \times 10^{-10}]$ mbar during both cleavage at 30 [60] K and measurements, the latter being carried out with a sample temperature of 10 [55] K. At the $1^3$ end-station, the samples were cleaved during the last stage of cool-down (T∼ 20 K, P∼ low $10^{-10}$ mbar), prior to insertion into the cryo-shielding of the $^3$He-cooled manipulator, and measurements were conducted at or just below 1 K. The overall energy resolution was set to 16(10) meV for the I05($1^3$) ARPES experiments and was roughly around 125 meV at I09. At I05, K getters (SAES) were used to deposit sub-monolayers of potassium atoms onto the cold, cleaved surface so as to shift the chemical potential into the (previously) unoccupied states. In addition, soft X-ray absorption at the Mn-$L_{2,3}$ edges was measured at I09 by recording the sample drain current. The photon energy resolution was set to 110 meV, and the X-ray absorption data were recorded at 55 K, at a grazing angle of incidence using linear horizontally polarised radiation.

## II. Theoretical procedures

First-principles density-functional-theory (DFT) calculations were performed using the all-electron full potential augmented-plane-wave method in the Wien2k implementation [21]. As discussed in the results and discussion section, we used two different exchange correlation potentials: the generalized gradient approximation (GGA) of the exchange-correlation potential [22], and the modified Becke and Johnson (MBJ) potential [23]. Below we report mainly the calculations for which the MBJ potential has been used, as these provided better agreement with the experimental ARPES data around the key $(0, \pi/a)$-point in $k$-space. Spin-orbit coupling is included as a perturbation, using the scalar-relativistic eigenfunctions of the valence states, a well-established way to implement spin-orbit coupling effect in most DFT codes such as Wien2k and VASP [24]. G-type antiferromagnetic order in the Mn planes was included in the DFT calculations, assuming a value for the local moment of 5 $\mu_B$/Mn atom, obtained by the self-consistent DFT formalism. Based on the DFT results, we optimised the parameters of a 6-band, tightbinding model, based on three $5p-$orbitals for each of the two inequivalent Sb atoms per unit cell. Details of the tight binding model are presented in Appendix I.

## 3  Results and discussion

### I. Crystal structure, magnetisation, resistivity and surface characterisation

The unit cell of $SrMnSb_2$, shown in Fig. 1(a), has an orthorhombic structure with the longest crystallographic axis denoted as $a$, and the planes of the layered structure spanning the $b$-$c$ directions. The unit cell parameters of $SrMnSb_2$ are $a = 23.011$, $b = 4.384$, $c = 4.434$ Å [15]. The space group is $Pnma$ (No. 62), which is a non-symmorphic group, featuring several screw axes and one glide-plane. There are two main groups of Sb atoms in the structure, labelled as Sb1 and Sb2 in Fig. 1(a). The Sb2 layers are tetrahedrally bonded to the Mn planes. The Sb1 atoms determine the main features of the near-$E_F$ electronic structure [15, 16], and are relatively weakly coupled to the Sr atoms, thus forming a quasi-2D layer. The staggered arrangement of the Sr atoms situated in chains above and below the Sb1 layer doubles the unit cell and introduces a screw axis along $b$, as shown in Fig. 1(b), and also results in modest out-of-plane buckling of the Sb1 layer. As a consequence, the Sb1 plane of the structure taken on its own - shown in the plan view in Fig. 1(b) - exhibits the key symmetry feature of the crystal: a quasi-2D, square network of heavy (Sb) atoms, possessing a non-symmorphic symmetry element. The moments of the Mn atoms are ferromagnetically ordered at high temperature, and form a staggered AFM arrangement below 304 K [15], whose magnetic unit cell is identical to the crystallographic unit cell.

Turning to the transport characterisation, both the temperature dependence and absolute value of the in-plane resistivity of our $SrMnSb_2$ crystals were very similar to that reported in Ref. [15]. The hole carrier concentration for our samples is between $1-1.5 \times 10^{19} cm^{-3}$, and the mobility (at 2 K) is high, at 15,000 $cm^2 V^{-1} s^{-1}$. Magnetic susceptibility measurements showed our crystals to possess a small magnetic moment of order 0.02 $\mu_B$/Mn atom. Comparing this to the detailed magnetic characterisation reported in Ref. [15], this would place our crystals in the intermediate moment regime of the three reported there.

Fig. 1(c) shows a typical LEED diffraction pattern from in-situ cleaved $SrMnSb_2$ recorded with a primary electron beam energy of 220 eV. There is a straightforward (1×1) pattern of spots, and no superstructures or reconstructions were seen on several different cleaves. Fig. 1(d) shows an STM topograph from a cleaved $SrMnSb_2$ crystal, with the (1×1) surface termination also clearly visible. The Sb2−Mn−Sb2 block has short interplanar bond lengths, suggesting strong covalent bonds, and thus the crystal would be unlikely to cleave inside this

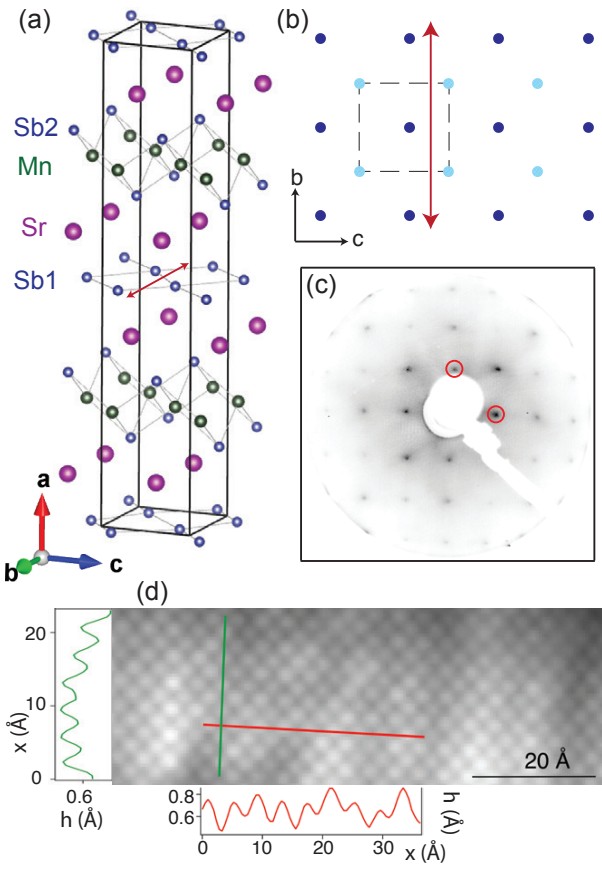

Figure 1: **Bulk and surface crystallographic structure.** (a): Crystal structure of SrMnSb$_2$, featuring two inequivalent Sb positions, labelled as Sb1 and Sb2. The Sb1 plane is sandwiched between two staggered Sr planes. The Sb2 atoms sandwich a plane of the Mn atoms. (b): A plan view of the Sb1 plane, the key structural element of the crystal. The buckling of the Sb1 plane out of the $b$-$c$ plane is indicated using different shades for the Sb1 atoms. A non-symmorphic symmetry element in the form of a screw axis is indicated as a red arrow aligned along $b$. (c): LEED diffraction pattern of an in-situ cleave of SrMnSb$_2$, recorded with an incident electron energy of 220 eV at 30 K. The pattern is of a (1×1) type, whereby the red circles highlight the (1,0) and (0,1) (h,k) diffraction spots. (d) STM topograph of cleaved SrMnSb$_2$ measured at 77 K in UHV using a Pt/Ir tip at $V = $ -600 mV and $I = 95.8$ pA. The same (1×1) surface symmetry is seen as in the LEED data, and the interatomic separation corresponds to the net of the Sb1 atoms.

structural unit. Angle-dependent analysis of the core level photoemission intensities of the different atoms in the structure (not shown) signalled that cleavage is most likely to take place in the Sr—Sb1—Sr unit, which is in line with the interatomic distances seen in the STM topograph. The latter match well with those of the Sb1 plane of the crystal structure. However, from symmetry considerations Sr termination is also possible, but we do not have any concrete experimental evidence for this.

The ARPES data that will be presented and discussed in detail below, showed no significant time dependences, remaining unchanged over the course of days in UHV. In previous ARPES experiments on 3D topological insulators such as $Bi_{2-x}Sb_xTe_{3-y}Se_y$ (BSTS) [25] or $YbB_6$ [26], we have seen clear time dependent changes in the Fermi level position, which are accepted to be caused by bandbending. However, we saw none of the tell-tale signs of downward or upward bandbending (and Fermi level shift) arising from the surface photovoltage and/or photon-stimulated desorption in the $SrMnSb_2$ ARPES data. In any case, the 3D carrier density of $SrMnSb_2$ of order of $10^{19}$ cm$^{-3}$ is three orders greater than that in the 3D topological insulator BSTS ($\sim 1 \times 10^{16}$ cm$^{-3}$ for x=0.58 and y=1.3 [27]) and thus $SrMnSb_2$ is simply too conducting/metallic for band bending to take place. Thus, all in all, the surface characterisation data point to a system enabling the generation of a stable and simple (1×1) cleavage surface.

## II. ARPES data

### (a) Overview of ARPES data

ARPES experiments carried out with VUV/EUV photon energies are highly surface sensitive, due to the short inelastic mean free path length of the photoelectrons in the solid, and are best carried out on crystals with the sort of 'well behaved' surfaces such as those described in the preceding section. The experiment provides direct access to the binding energy of the electronic states (referenced to the Fermi level as zero), as a function of their crystal momentum parallel to the surface of the sample, and thus give a window on the occupied band structure, as well on the electronic self energy of the system. For simplicity, we will refer to the $k$-components of the crystallographic $b-c$ plane as $k_x$ and $k_y$. Due to the surface sensitivity, there is a degree of integration over $k_z$, and thus ARPES data are generally the most clear and sharp from systems with electronic states whose energies depend primarily on $k_{x,y}$: in other words quasi-2D electronic systems.

Fig. 2 presents a compilation of ARPES data from $SrMnSb_2$, recorded with various photon energies on three different beamlines. Panel (a) shows a large ARPES 'data block' measured using 88 eV photons. The cutaways are placed so as to reveal the dispersion relation ($E(k_{x,y})$) along different high symmetry directions in reciprocal space, which are labelled corresponding to the $k_z$=0 plane of the Brillouin zone shown in the inset. Below a binding energy of 0.6 eV, the ARPES data show $SrMnSb_2$ to have a simple band structure with two main dispersive sets of bands. The key feature in the data are the very sharp, linearly dispersive bands that look to be approaching closure to a point, centred at the Y and X points in Fig. 2(b). The sharpness of these structures suggests that these states may be quasi-2D in nature (something we return to below), in keeping with the layered crystal structure and the ratio of ~600 (at 2 K) between out-of-plane and in-plane components of the resistivity reported in Ref. [15].

There is also a second, broader feature in the electronic structure. This band's dispersion is less steep, and it yields a larger, more roughly contour centred on Γ with a reasonably well defined outer perimeter, but significant 'filling-in' inside the contour. One recipe for this kind of behavior in ARPES data would be a hole-like parabola, with $E_F$ situated close to the top of the parabola, such that the bands are starting to bend over, providing intensity in the constant energy contour across a range of $k_{x,y}$ values. The ARPES data presented in Fig. A2 of Appendix

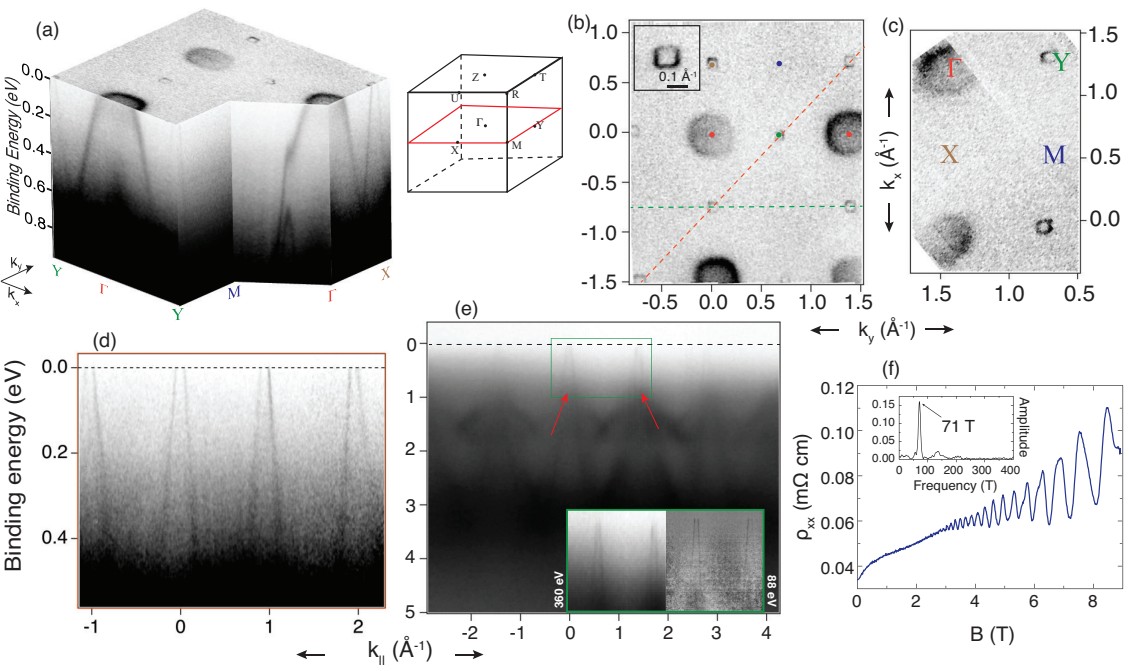

Figure 2: **ARPES data overview.** (a) ARPES data block (I05, h$\nu$=88 eV, circular right polarisation, T=10 K) showing the electronic structure of SrMnSb$_2$ vs. the in-plane momentum, $k_{x,y}$. High symmetry points for the surface projected Brillouin zone are labelled, and a schematic of the bulk and surface-projected Brillouin zones is included. The top surface displays the Fermi surface of the system, which is shown in a plan-view in panel (b), including a zoom of the small Fermi surface, recorded using 27 eV photons. The contour of the Fermi surface centred at the Y-point ($k_x$=0, $k_y$=-0.7 Å$^{-1}$) is a rounded square, with an area of only 0.3% of the 2D Brillouin zone. Panel (c) shows a Fermi surface map recorded at BESSY-II (h$\nu$=95 eV, circular right polarisation, T=1 K) in which only a single twin domain of the sample is picked up in the measurement, showing clearly the small FS's are now only centred on the Y-points. Panel (d) shows a cut taken along the orange dashed line in panel (b), which dissects four of the small Fermi surfaces, highlighting their very straight dispersion relation. Panel (e) contains a different cut through $k$-space, showing a line through either X or Y-points alternating with M-points, measured using soft X-rays (I09, h$\nu$=360 eV, linear horizontal). The inset superimposes an analogous cut measured using 88 eV photons, as in panels (a), (b) and (d), showing that relaxing the extreme surface sensitivity of the VUV excitation does not alter the electronic structure. Panel (f) shows the very clear quantum oscillations observed in the resistivity (SdH) from crystals from the same growth batch. The Fourier transform in the inset shows a fundamental frequency of 71 T, which would correspond to a Fermi surface area of 0.33% of the 2D Brillouin zone, in excellent agreement with the ARPES data for the Y-centred Fermi surfaces.

II show that this is, indeed, the case here.

Fig. 2(b) shows that the Fermi surface of $SrMnSb_2$ comprises of small features centred at Y and X (from the high-velocity, linearly dispersing bands), and the larger, broader $\Gamma$-centred features. The inset shows a zoom of the small Fermi surface (FS), recorded with lower photon energy (27 eV) to provide improved $k$-resolution. The area of this small FS pocket is found to be only 0.3% of the 2D cut through the bulk Brillouin zone. We note that despite the small size of the Y-FS, its centre is empty and its energy contour is sharp in $k$-space. This is clear evidence that only a single band or a set of perfectly degenerate bands cuts $E_F$ at this $k$-location. We note here that the $b$ and $c$ lattice constants in this orthorhombic system are the same within $\sim 1\%$ [15], and thus the small Y-centred Fermi surface appears 4-fold symmetric.

Fig. 2(c) shows an ARPES FS map recorded at the $1^3$ end-station at BESSY-II, which is qualitatively different to the data from the cleaves studied at Diamond Light Source's I05 or I09 beamlines. It is clear that the X-centred FS's are missing from these data, meaning that the Brillouin zone will contain a single FS centred at Y. As we will see later when discussing the theory results, this is, in fact, expected for the band structure of $SrMnSb_2$. The reason for the difference between the data of Figs. 2(a) and 2(b) is that in the former, the synchrotron light spot was averaging over multiple twin domains of the orthorhombic crystal, meaning the data are a sum of the electronic states from sets of twin domains in which the orthorhombic $b$-$c$ planes of $SrMnSb_2$ are rotated by 90° with respect to each other.[1] Serendipitously, the data shown in Fig. 2(c) were recorded from a single domain.

Panel (d) of Fig. 2 shows a perpendicular cut through the 3D data block, going through the X—Y—X points, as indicated by the orange dashed line in Fig. 2(b). This direction in $k$-space highlights the sharp states which give rise to the small Fermi surfaces — the Y-states — and these remain very sharp up to a binding energy of 400 meV and look to possess a linear dispersion relation.

In order to probe the electronic structure deeper into the crystal, in Fig. 2(e) we show data recorded at Diamond Light Source's I09 using soft X-rays with a photon energy of 360 eV. A cut through the Y[X]—M—Y[X] direction is shown, indicated by the green dashed line in panel (b). The superimposed ARPES data recorded using 88 eV photons show that the bands for VUV and SX-ARPES resemble each other strongly, even though the inelastic mean-free path length for 360 eV electrons is double that for excitation at 88 eV [28]. The fact that modifying the surface sensitivity of the ARPES experiment leaves the data essentially unchanged, points to an absence of surface-related electronic states that are different to those of the bulk of the crystal. In addition, the main panel of Fig. 2(e) shows that soft X-ray excitation leads to better definition of the features with binding energy exceeding 0.4 eV, allowing the states to be tracked to a band bottom at M at a binding energy close to 4 eV, a value that is used to control the parameters of the tight binding model presented later in the paper.

The last data panel (f) of Fig. 2 is a crucial one, as it presents magnetotransport data enabling a direct comparison of surface sensitive spectroscopic (ARPES) and bulk sensitive transport (SdH) data from crystals from the same source and growth batch. Shown is the resistivity data measured at 2 K vs. applied (out-of-plane) magnetic field. It is notable that already from fields as low as 3 T, clear quantum oscillations are evident, attesting to the high mobility of the carriers giving rise to these oscillations. The inset shows the FFT of the magnetoresistance data which gives the frequency, $F$, of these oscillations as 71 T, similar to the values reported previously [15]. The Onsager relation, $F = (\Phi_0/2\pi^2)A$, enables the area of the Fermi surface pocket ($A$) to be extracted from $F$, and these data give a value of 0.33% of

---

[1]Since the BESSY-II data captured a single domain, this shows domains can be larger than the spot size of ca. 40x200 $\mu$m. We do not expect the small number of domain walls that one could consequently expect across the mm-sized area of the sample on which transport is done to significantly influence the quantum transport phenomena being analysed.

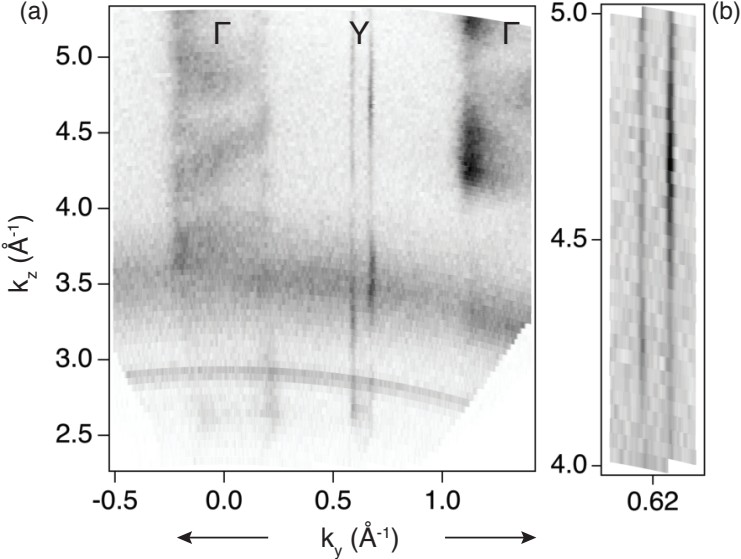

Figure 3: $k_z$-**dependence of the electronic states.** (a) $k_z$ vs $k_y$ intensity map for E=E$_F$ for a Γ−Y cut, obtained from photon energy dependent ARPES experiments (h$\nu$=20-120 eV). (b) Zoom-in on the Y state over a $k_z$ region covering almost four Brillouin zones along $k_z$. It is clear that $k_F$ for the Y-state shows little or no dependence on $k_z$, implying the quasi-2D nature of these states.

the area of a 2D cut through the Brillouin zone (with an error bar of order ±5% of this value extracted from the fit errors to the Fourier transform of the QO data). This is in excellent agreement with the area of the Y-state Fermi surface pocket observed in the ARPES data of panels (a-c) of Fig. 2.

This means that SrMnSb$_2$ is a material for which the electronic structure of the outermost nanometer of the crystal (from ARPES) agrees quantitatively with the results of bulk sensitive resistivity measurements, and importantly, this enables us to unambiguously link the Y-states to the quantum oscillation data in the magnetotransport in high quality single crystals of this material. This means that the broader states seen crossing $E_F$ close to the Γ-point in ARPES play little role for the transport physics of SrMnSb$_2$, and it can be speculated that their mobility may be much lower. Relevant to this last point is the fact that we show in Appendix II that an upward shift in the chemical potential due to adsorption of K atoms on the cleavage surface of SrMnSb$_2$, places E$_F$ above these Γ-centred states (i.e. they are absent from the Fermi surface), but leaves the shape of the Y-centred Fermi surface unmodified.

**(b) ARPES: $k_z$ dependence of the electronic states**

Given the $k_z$-integration mentioned earlier in the context of the short inelastic mean-free path length for regular ARPES, the sharpness of the Y-centred small FS's in these data could be taken to imply the 2D character of the electronic states involved. In order to confirm this, we measured the $k_z$ dependence of these states by varying the incident photon energy from 20 to 100 eV. Fig. 3(a) shows an ARPES intensity map for E=E$_F$ for a $k_z$ vs $k_y$ plane, with $k_y$ along Γ−Y. None of the ARPES features show much variation with photon energy, except for some intensity variations inside the Γ state. Although the latter could be due to a degree of 3D nature for the Γ states which can be seen in the DFT data of Fig. 5(a), the variations observed in the h$\nu$ dependent ARPES data do not possess the required periodicity in $k_z$, and thus are probably heavily impacted by $k_z$-dependent changes in the photoionisation cross section and

matrix elements. The long unit cell dimension along the out-of-plane direction means that a single Brillouin zone is only 0.27 Å$^{-1}$ in $k_z$. Therefore, any truly 3D bulk state would have to show significant curvature along the $k_y$ direction repeating within $k_z$ intervals of $|0.27|$ Å$^{-1}$ in such an ARPES map. A zoom-in of the Y state is shown in Fig. 3(b), covering a $k_z$-region spanning almost four Brillouin zones. It is very clear that the Y-state FS does not close at any $k_z$, indicating the 2D, or quasi-2D nature of this state, which our theory analysis will show has its origins in the 2D Sb1 plane in the crystal structure of SrMnSb$_2$. The data unequivocally underpin an 'open' FS vs. $k_z$, and although we cannot exclude a periodic corrugation in the $k_F$ value along $k_y$ as a function of $k_z$, its level cannot exceed 15% of $k_F$, implying that the electrons in these pockets can very well be approximated as almost ideal 2D states.

### (c) ARPES: fit of dispersion and self energy of the Y-states

Given the relevance of the Y-states to the bulk electrical transport properties of SrMnSb$_2$, a more quantitative analysis was performed. The standard method of fitting the momentum distribution curves (MDC's) with Lorentzian curves for the left and right-moving states was employed, and the data and fit results are shown in Fig. 4. For systems with linear dispersion, the FWHM of the Lorentzian fit is $\Delta k(\omega) = 2\Sigma''(\omega)/v_F^0$, where $\Sigma''(\omega)$ is the imaginary part of complex self-energy and $v_F^0$ is the bare Fermi velocity [29]. Fig. 4(a) shows the underlying ARPES data along $\Gamma$–Y, with the fit positions shown as blue dots on the left-hand branch, and the extracted MDC widths overlaid as red bars on the right-hand branch. The orange dotted line is an extrapolation of the extracted dispersion relation, which looks to also match the data at greater binding energies than used in the fit. Panel (b) shows the fitted MDC peak widths, and panel (c) a linear fit to the fitted MDC positions, yielding a velocity of 5.5 eVÅ, or 8.4×10$^5$ m/s. The MDC widths for the Y state can be seen to increase from 20 mÅ$^{-1}$ at E$_F$ to 30 mÅ$^{-1}$ at a binding energy of 100 meV. Such narrow MDC lineshapes close to E$_F$ have been recorded from topological surface states in systems such as Bi$_2$Se$_3$, Bi$_2$Te$_3$ and for the Dirac cone dispersion of graphene [30–33]. Such sharp MDC features have also been seen in the superconducting state of Bi$_2$Sr$_2$CaCu$_2$O$_8$, which is also a highly 2D system [29]. In terms of the energy dependent growth in the MDC width, SrMnSb$_2$ shows faster broadening than the surfaces states of the 3D TI's but is slower than the case of BSCCO. Finally, we note that if one permits an extrapolation of the ARPES data of the Y-states to energies above E$_F$, Fig. 4(a) indicates the putative crossing point of the linear bands would lie at around 200 meV above the Fermi level.

At this stage we pause to take stock of the information coming thus far from these first ARPES data on SrMnSb$_2$. This material is proposed to be a topologically non-trivial system [15]. The ARPES data show Y-states to have linear dispersion, and to be very sharp. There is also a perfect match between the FS formed by the Y-states in ARPES and that from SdH-QO experiments. Were this paper to stop here, the conclusion could easily be that SrMnSb$_2$ is indeed a realisation of a 3D crystal harbouring gapless 2D Dirac fermion states, with the only complication being that E$_{Dirac}$ is located 200 meV above E$_F$.

## III. DFT and TB analyses

### (i) Density functional theory results

Instead of stopping there, we continue with a detailed theoretical analysis of the electronic states of SrMnSb$_2$ using both density functional and tight binding calculations. According to published DFT calculations, SrMnSb$_2$ is an insulator with a large band gap around the Y-point of $k$-space [16]. This clearly presents a poor match with what is seen in ARPES and in transport. Since it is well-known that the choice of exchange potential may significantly influence the size

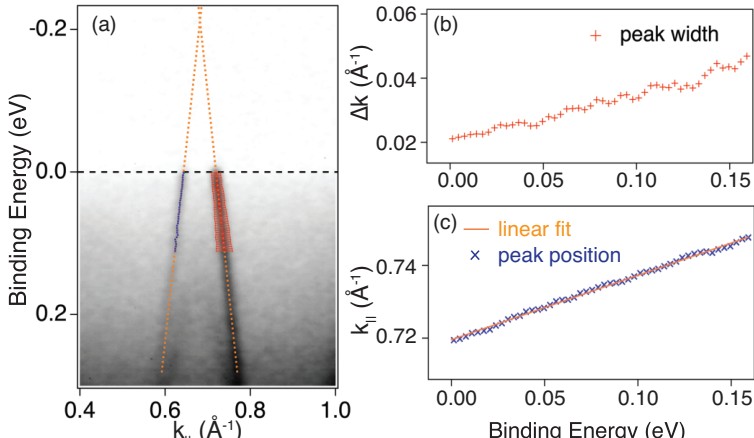

Figure 4: **Self energy analysis of the Y-states** (a) $I(E,k_{||})$ along $\Gamma-Y$ measured with 88 eV photons. The data are overlaid with blue dots (MDC peak position from Lorentzian fit) and red bars (fitted MDC width) and the orange dotted line shows the fitted dispersion relation. (b): MDC peak widths $\Delta k$. (c) A linear fit to the dispersion matches well with the fitted MDC peak positions.

of the band gaps in DFT predictions, we compared predictions of our own DFT calculations using various exchange potentials with our ARPES data. We found similar results to Ref. [16] when using the generalized gradient approximation (GGA) for the exchange correlation potential. However, when doing the DFT calculation using a MBJ (modified Becke-Johnson) exchange potential, which provides a better description of band gaps in semiconductors and (correlated antiferromagnetic) insulators [23, 34], and including the effects of spin-orbit coupling, we get significantly different results, which we summarise in Fig. 5(a). Like the ARPES data, the DFT results also show two main groups of bands close to $E_F$: hole-like states centred on $\Gamma$, which just touch the Fermi energy in the calculations, and high-velocity bands centred on the Y-point that get very close to $E_F$, but — significantly — *do not* cross the Fermi level and thus display a gap.

Concentrating on the occupied states accessible to ARPES, and accepting that the chemical potential in a real crystal does not agree with the DFT expectation, we show in Fig. 6 that the MBJ-based DFT results — when shifted upwards by 66 meV — match very well to the measured ARPES data around the $\Gamma$-point and in their description of the high-velocity bands centred on the Y-point. What the DFT reveals is that the Y-centred states possess a gap of ca. 200 meV above the Fermi level relevant for the experimental data. We note here that the optical conductivity data from SrMnSb$_2$ [35], indicate the presence of an electronic transition across a gap of this order.

The DFT data show the states close to $E_F$ to be independent of $k_z$ between the Y and T points (see the inset to Fig. 2(a) for the Brillouin zone labels), indicating the 2D nature of the Y-states as shown in the ARPES data of Fig. 3. The DFT data of Fig. 5(a) show that two-thirds along both the Y$-\Gamma$ ($k_z$=0) and T$-$Z ($k_z=\pi$) $k_{x,y}$-directions there is no significant lifting of the degeneracy of the two bands involved. If this also applies to $k_z$ values between these extrema, then this can explain the 'straight-up' edges seen in the $k_z$ dependent ARPES data at $\pm$ 0.2 Å$^{-1}$ either side of $\Gamma$ in Fig. 3(a). However, for $k_{x,y} = 0$ the DFT data of Fig. 5(a) do show dispersion along $k_z$ (between $\Gamma$ and Z), with the two bands being degenerate at Z and not so at $\Gamma$.

The DFT data are very clear that the high velocity states only cross $E_F$ at the Y-point, and not at the X-point, a fact that enables us to understand the differences between the ARPES FS maps

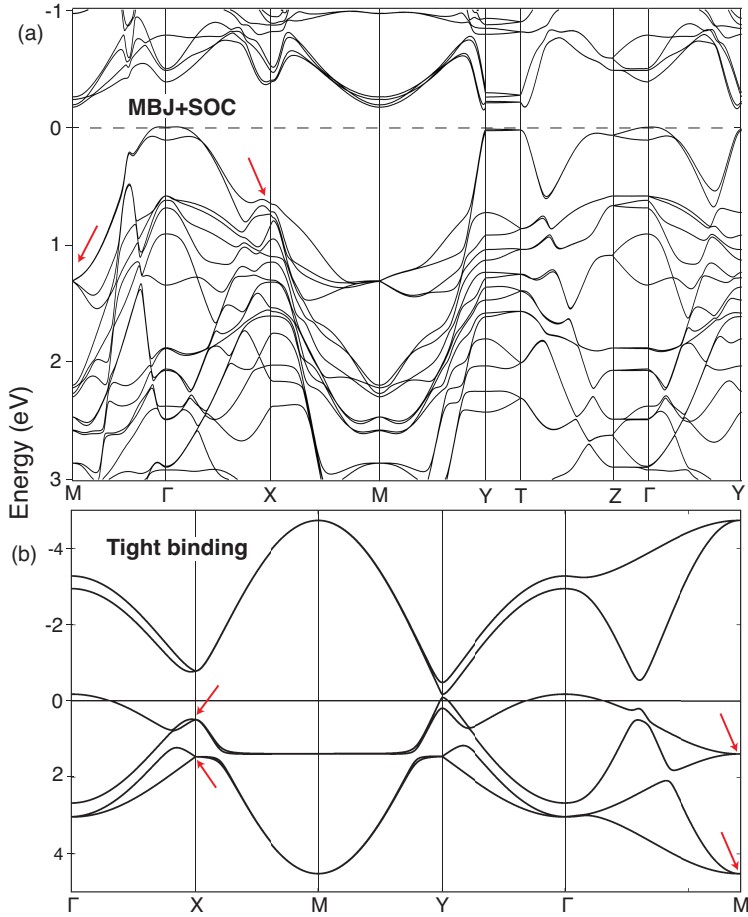

Figure 5: **Electronic structure of SrMnSb$_2$ from DFT and tight binding calculations** (a) Calculated band structure of SrMnSb$_2$ using a MBJ exchange potential within the DFT formalism. Intrinsic spin-orbit coupling and magnetism are also included; (b) calculated band structure from a tight binding model for the $p_{x,y,z}$ orbitals of Sb atoms arranged in a buckled square lattice as is the case in the SrMnSb$_2$ structure. In both cases, the red arrows point to band crossings at X and M that are protected by non-symmorphic symmetry.

shown in panels (b) and (c) of Fig. 2. The difference between the electronic states around the X- and Y-points in the DFT (and TB) data has its origin in a small displacement of one subset of the Sb1 atoms along the crystallographic (in-plane) $c$ direction (a displacement that in fact removes a second in-plane screw axis that would otherwise run along the $c$-direction).

Summarising the main result of the DFT calculations — which are validated by their excellent agreement with the experimental electronic structure of SrMnSb$_2$ — we can state that the high-velocity states around the Y-point that are very sharp in ARPES and look to have linear dispersion *do not* extrapolate to gapless band crossings which are protected by any lattice symmetry that is present in the calculations. This fact already places SrMnSb$_2$ outside the Dirac family. Whether these gapped states at Y can nonetheless possess non-trivial topology is a point we return to later.

**(ii) Results of tight binding calculations**

In order to gain more insight into this conclusion and to examine whether there are non-symmorphic protected crossings in SrMnSb$_2$ elsewhere in the band structure (away from the Y-point and/or away from E$_F$), we set up a simple tight binding (TB) hamiltonian for the $p_{x,y,z}$

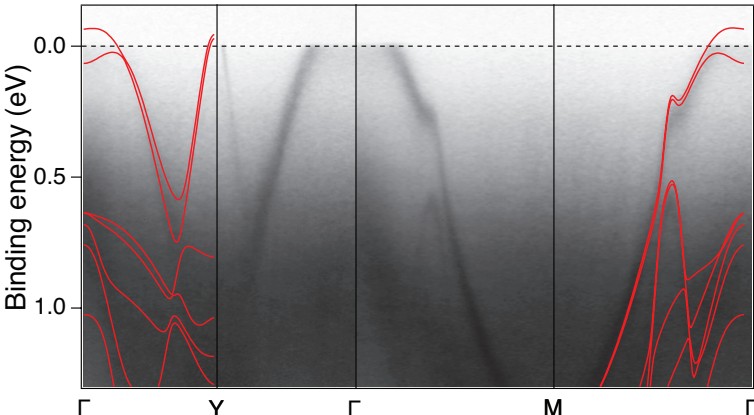

**Figure 6: Excellent agreement between DFT (MBJ potential) and ARPES data**. ARPES data along Γ—Y—Γ—M—Γ directions in *k* space, with the DFT bands overlaid in each case. The Fermi energy position in the DFT data has been shifted downward by 66 meV, compared to the original ab initio data shown in Fig.5(a).

orbitals of the Sb atoms in the Sb1 plane (see Appendix I for details). The results of the tight binding band structure including atomic spin-orbit coupling as well as the second neighbour spin-orbit interaction, made possible by the buckling of the Sb1 plane (which is equivalent to the $t^{SO}$ term in Young and Kane's model presented in [17]), are shown in Fig. 5(b). The TB parameters are chosen so as to yield good agreement both with the DFT as well as with the ARPES data, exploiting the extended binding energy range over which the bands can be tracked in the soft X-ray ARPES data such as those shown in Fig. 2(e).

Our TB calculation shows that there *are* degenerate points (band crossings) protected by non-symmorphic symmetry in SrMnSb$_2$. In agreement with the predictions by Young and Kane [17], these are located at the X- and M-points, as indicated by red arrows in both panels of Fig. 5. In the DFT data, the strong (5 $\mu_B$/Mn atom) magnetism assumed in the calculation impacts the electronic states at these $k$-locations, as will be discussed in more detail in Section VI.

Comparing the DFT and TB calculations with the SX-ARPES cut shown in Fig. 2(e), the two maxima at the Y/X-points that can be seen at E$_F$ can be assigned to the states at the Y-point, and the maxima seen at a little under 1 eV binding energy in the ARPES data (the region of which is indicated using a red arrow in Fig. 2(e)), to the groups of bands with a maximum around the X-point in the theory data of Fig. 5. These are in the right region of (E,$k$)-space to contain one of the special, non-symmorphic protected crossings at the X-point (red arrows in Fig. 5), but obviously the level of distinction in the ARPES data precludes their explicit identification. Furthermore, the two minima visible in the SX-ARPES data at the M-points at ca. 1.5-1.8 and 3.6 eV below E$_F$ clearly match with the two sets of bands showing band bottoms at M in the DFT and TB calculations.

Completing our discussion of possible symmetry-protected crossings, we note that regardless of the role played by the magnetism on the Mn lattice, the theoretical data show that the band filling in SrMnSb$_2$ puts the crossings protected by non-symmorphic symmetry at 0.7 eV below E$_F$, and thus they play no role in the transport properties of SrMnSb$_2$.

## IV. Berry phase for the Fermi surface at the Y-point

From the combination of the ARPES and quantum oscillation data it is clear that the electronic states crossing E$_F$ at the Y-point are responsible for the electronic transport in SrMnSb$_2$. The theory analysis is equally clear that these same states do not enjoy protection by the non-

symmorphic space group symmetry and are gapped albeit above the Fermi level. A remaining question is whether these states — even though gapped — possess non-trivial topology, i.e. whether they accumulate a Berry phase of $\pi$ upon going around the Fermi pocket. This question is all the more relevant considering that a non-trivial Berry phase has been reported from quantum oscillation data from SrMnSb$_2$ in Ref. [15]. We first approach this issue from the point of view of theory.

In Fig. A1 (in the appendix) we illustrate how our tight binding approach is able to capture the essence of the occupied bands around the Y-point by comparing with constant energy contours from the ARPES data. On the basis of this good agreement with both the ARPES and DFT data, an analysis of the Berry curvature for the states around Y has been made using the TB results.

In order to arrive at dependable measures of the Berry phase from a numerical calculation, it is important that the method used is gauge invariant [36]. In our case, we use the TB Hamiltonian, $\mathbf{H}(k)$ to define the Berry phase $\gamma_n$ in terms of Berry curvature $\mathbf{\Omega}_n(k)$ as:

$$\gamma_n = \int_S d\mathbf{S} \cdot \mathbf{\Omega}_n(k), \tag{1}$$

in which

$$\mathbf{\Omega}_n(k) = \mathrm{Im} \sum_{m \neq n} \frac{\langle n(k)|\nabla_k \mathbf{H}(k)|m(k)\rangle \times (m \leftrightarrow n)}{(E_m(k) - E_n(k))^2}, \tag{2}$$

where $n(k)$ are the eigenstates of $\mathbf{H}(k)$.

In the TB data (with time reversal and inversion symmetry), the hole-like band at Y is comprised of two perfectly degenerate states, and the Berry curvature analysis shows the combination of these two bands to have zero Berry curvature over the entire area enclosed by the Fermi pocket. We have checked that this situation is valid for situations in which the chemical potential is lying a little deeper or shallower in the band structure (like the constant energy contours shown in Fig. A1 for E = 50 or -100 meV binding energy). This range easily covers all the variation in the chemical potential (and thus the area of the Y-state Fermi surface, expressed in terms of the frequency seen in the Fourier transform of the Shubnikov de Haas oscillations) reported either here or in Ref. [15]. Thus the TB analysis signals that the states around the Y-point, which we know from the comparison between the ARPES and magnetotransport QO data are responsible for the transport in SrMnSb$_2$, are topologically trivial.

In reality, SrMnSb$_2$ is known to be a canted antiferromagnet at low temperature. Can this additional breaking of time reversal symmetry lead to non-trivial topology around the Y-point? The Sb 5p-related states giving rise to the Fermi surface at Y are in an internal magnetic field due to the canted AFM Mn moments, although the Sb1 plane is quite far from the Mn planes in the crystal structure. To take this into account we have added the effect of a magnetic field to the TB analysis. On a principal level the magnetism lifts the two-fold degeneracy of the bands at Y. However, even taking significantly larger values of the local magnetic field than those expected in SrMnSb$_2$, the splitting we obtain at the Y point is never larger than 10 meV at most. As a result, even with a very fine tuning of the chemical potential so that it intersects only one of the non-degenerate bands at Y, the TB data show the integrated Berry curvature enclosed by the Y Fermi pocket never exceeds 10% of $\pi$, thus leaving the theory statement essentially unchanged that the states at Y are topologically trivial. An independent check within our DFT calculation in which G-type AFM is included also shows a Berry phase of zero upon circling the Y-point. We pick up the point of the Berry phase again below when discussing the quantum oscillation data from our crystals in the following section.

## V. Berry phase analysis of the transport data

The field dependence of the Landau levels (LL) in a SdH quantum oscillation experiments offers the possibility to test whether the Fermi surface of a material is comprised of topologically trivial or non-trivial states. In particular, plotting the LL indices vs. 1/B, and extrapolating the resulting straight line fit to infinite magnetic field — i.e. determining where the extrapolation of 1/B→ 0 cuts the LL index axis — yields a quantity labelled $n_0$. If $n_0$ is zero, then the system is topologically trivial, if it is ±0.5 it is topologically non-trivial.

Fig. 7(a) shows the longitudinal and Hall resistivities for SrMnSb$_2$ plotted vs. 1/B. In the inset to panel (b) of Fig. 7, we show the oscillations in $\sigma_{xx} = \rho_{xx}/(\rho_{xx}^2 + \rho_{xy}^2)$, calculated from the data in panel (a). The main part in Fig. 7(b) shows the Landau level index plot from these conductivity oscillations. It is clear from panel (a) that for fields below 4T (i.e. 1/B > 0.25), the $\Delta\rho_{xy}$ (red) curve does not display oscillations. This, in fact, precludes accurate extraction of $\Delta\sigma_{xx}$ beyond 1/B > 0.25 T$^{-1}$. For this reason we limit the linear fit to the LL indices to the data points shown in panel (b) and this yields an $n_0$ value of 0.14. If we repeat the fit to the LL-index plot but include *all* the data in Fig. 7(a) - i.e. also for 1/B > 0.25 - then $n_0$ becomes -0.16.

It is known that slight corrugation of a quasi-2D Fermi surface in $k_z$ is able to yield a non-zero axis intercept, giving $n_0$ values of ±1/8 even for topologically trivial states [37]. The degree of $k_z$-corrugation required would be in keeping with the upper limit of 15% set on the possible $k_z$-dependence of the Fermi surface at the Y-point discussed in the context of the photon energy dependent ARPES data.

Corrugated or not, ARPES data show that the small Fermi surface at the Y-point in $k$-space is responsible for the transport in our crystals of SrMnSb$_2$, and the QO data from the same states in the same crystals attests to their topologically trivial nature in our samples, a conclusion in line with the TB theory analysis discussed in the previous section.

It is pertinent to note here that the data from some of the samples published in Ref. [15] support a different conclusion, namely that the states are topologically non-trivial. The samples whose data are consistent with a $\pi$ Berry phase have: (i) a smaller Y-state Fermi surface area in the QO data (FFT frequency of 67-69 T, rather than 72 T, or 71 T in our case) and (ii) larger magnetisation (M > 0.1 $\mu_B$/Mn atom) [15] than those we have grown and studied here (M ~0.02 $\mu_B$/Mn atom).

Given the energy vs. $k_x, k_y$ information our ARPES data gives us, we can assess that the maximal variation in SdH oscillation frequency (69 vs. 72 T) would correspond to a chemical potential lying ~10 meV above where it does in our SrMnSb$_2$ crystals. The stronger magnetism in the crystals with $\pi$ Berry phase in Ref. [15] could point to an important role, for the time reversal symmetry violation lifts the two-fold degeneracy of the states that we know form the Y-centred Fermi surface pocket. Here we return to the discussion initiated in the previous section as regards to the impact of magnetism on the electronic states of the Sb(1) plane. From the TB analysis, it is clear that given TRS the states at Y are two-fold degenerate. The issue then boils down to whether the splitting resulting from the observed magnetism — perhaps in conjunction with a fortuitous positioning of the chemical potential in only the upper of the two states — could give rise to a non-zero Berry phase for certain samples. The very small splitting expected from the TB model would suggest this is unlikely. However, the close-to-$\pi$ Berry phase numbers from the QO for some of the samples in Ref. [15] perhaps point to a more complex reaction of the electronic states to the magnetism in the system than can be theoretically explained by the TB calculations at this time.

The TB and DFT data, however, do make the point clear that the electronic states around Y do not enjoy protection from gapping and thus SrMnSb$_2$ certainly is not a robust Dirac system, with special invariant characteristics regardless of the details of the defect chemistry or stoichiometry of each individual sample. Indeed, none of our samples showed any signs of

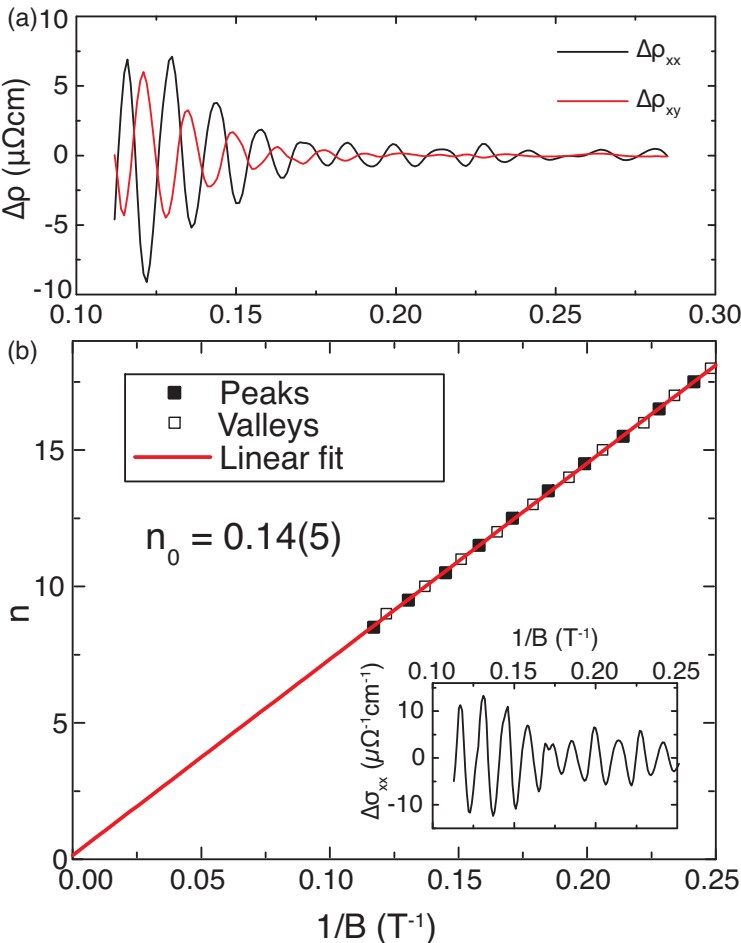

Figure 7: **Landau level diagramme for the analysis of the Berry phase in SrMnSb$_2$** (a) $\rho_{xx}$ and $\rho_{xy}$ vs. 1/B after background subtraction; (b) Landau level index diagram for the SdH oscillations in the conductivity, plotted by assigning integer values to the valleys of $\sigma_{xx} = \rho_{xx}/(\rho_{xx}^2 + \rho_{xy}^2)$ — shown in the inset. The linear fit gives n$_0$ = 0.14, which indicates a trivial Berry phase.

topologically non-trivial behaviour in the QO experiments.

## VI. Mn-L$_{2,3}$ soft X-ray absorption data, and role of magnetism in SrMnSb$_2$

Electrical transport in this material has been shown above to take place via states located at the Y-point in $k$-space, which are strongly two-dimensional, possess high mobility and show linear dispersion, yet are of trivial topology. As mentioned earlier, the crystal structure of SrMnSb$_2$ does enable non-symmorphic symmetries to protect band crossings at the X- and M-points in the band structure, but these states are well below the Fermi level.

In this section, we discuss the role played by magnetism in SrMnSb$_2$, detailed experimental data on which are reported in Ref. [15]. A canted AFM state was found with the effective FM moment aligned along the in-plane $b$-direction for temperatures below 304 K. In Fig. 8, soft X-ray Mn−L$_{2,3}$ absorption data of SrMnSb$_2$ are presented, and compared with data from Mn doped gallium nitride, in which the Mn has a tetrahedral crystal field similar to SrMnSb$_2$, and is known to be divalent [38], possessing the 3d$^5$ configuration.

Although the details of the electronic structure of the dilute magnetic semiconductor and SrMnSb$_2$ differ, the resemblance of the two soft X-ray absorption spectra represent a strong

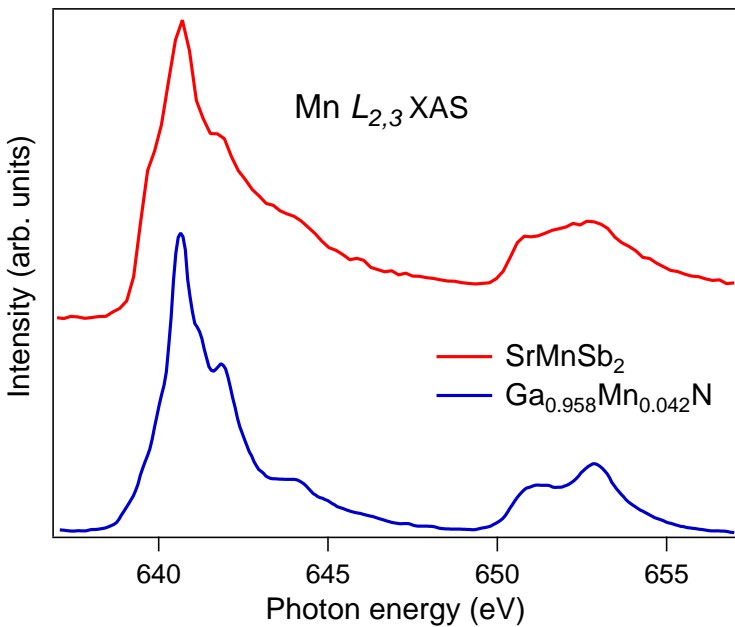

Figure 8: **Soft X-ray absorption at the Mn−L$_{2,3}$ absorption edge of SrMnSb$_2$**. The upper trace shows the spectrum of transitions from Mn2p→3d orbitals for a cleaved crystal of SrMnSb$_2$ measured at grazing incidence at the I09 beamline of Diamond Light Source at 55K with s-polarised X-rays. The lower trace shows data taken from Ref. [38] for Ga$_{0.958}$Mn$_{0.042}$N, which was modelled as being 3d$^5$, with a (tetrahedral) crystal field of -0.5 eV [38].

argument for Mn being in a 3d$^5$, high spin configuration in SrMnSb$_2$. The DFT calculations were carried out with fully spin polarised Mn 3d states carrying 5 $\mu_B$/Mn atom, and this choice is therefore backed up by the data of Fig. 8.

Here we would like to discuss the effect of the magnetism on the band crossings protected by non-symmorphic symmetry at the X- and M-points of SrMnSb$_2$ (see red arrows in Fig. 5). In the tight binding calculations, starting from a situation of protected crossings comprising of four bands (which can be thought of as two superimposed Dirac cones), addition of a magnetic moment gaps out the states. These gaps are small in the TB data, becoming clearly visible only for greatly exaggerated magnetic moments. Nevertheless, the principle point is that the magnetism gaps out the protected crossings in the TB calculations. Zooming in around the states at the X-point in the DFT data, for which the crossing is protected in the absence of magnetism, also shows that the internal field leads to gapping and non-degenerate states.

The magnetic order at low temperature in SrMnSb$_2$ is three dimensional in nature, and although at best only weakly dependent of $k_z$, the electronic states of the Sb1 planes are still periodic in three $k$-dimensions. If the system were to be a true Dirac system with a ferromagnetic ground state, then the breaking of TRS should lead to the formation of a Weyl fermion system, and the appearance of Fermi arcs as 2D edge states connecting the projections of the sources/sinks of Berry phase in the 3D Brillouin zone, as has been seen using ARPES for YbMnBi$_2$ [13]. It is very evident from the ARPES data shown in Fig. 2 that the experimental data on the electronic structure at the surface of SrMnSb$_2$ offers no room for the existence of Fermi arcs. This inability to form a Weyl state despite the effective ferromagnetism in SrMnSb$_2$ means either the system is perfectly 2D, or not a Dirac system, or both.

# 4 Conclusions

We have presented an investigation of SrMnSb$_2$ combining surface (ARPES) and bulk (transport) probes of the electronic states and an in-depth theory analysis. This material belongs to a much-discussed family of materials in which topological transport, Dirac semi-metallic behaviour and Weyl physics have been proposed recently. In addition, the non-symmorphic symmetry present in this system and the quasi-2D crystal structure have raised the question whether the Sb1 plane in this material is, in fact, the first realisation of a 2D Dirac semimetal in a 3D crystal, with Dirac band crossings protected against the effects of strong spin orbit coupling.

To the best of our knowledge, these are the first ARPES data from SrMnSb$_2$. Our data have shown that reconstruction-free and time-stable (001) cleavage surfaces of SrMnSb$_2$ can be generated, exposing the Sb1 crystal plane and enabling the collection of high-resolution ARPES data of very high definition and quality. Results from bulk probes of the electronic structure and theory agree quantitatively with the ARPES data. The key elements of the electronic structure emerging from a combination of the experimental data and theory are:

- High-velocity, very sharp, linearly dispersing, two-dimensional states that cross E$_F$ yielding small Fermi surface pockets centred at Y in $k$-space.

- Broader states centred on the Γ-point that drop below E$_F$ on a slight raising of the chemical potential (see Appendix II).

- Comparison between Shubnikov de Haas and ARPES data from single crystals from the same source and growth batch show unequivocally that it is the small FS pockets at Y that carry the current in SrMnSb$_2$.

- Both TB and DFT theory analyses, as well as analysis of the Landau level index diagram from our own SdH-QO data lead to a clear verdict that the electronic states around the Y-point at and close to the Fermi level in our samples of SrMnSb$_2$ are topologically trivial.

- Four-fold band crossings exist in SrMnSb$_2$ protected from gapping (in the absence of magnetism) by non-symmorphic symmetry at the X- and M-points. These states lie well below the Fermi level, and are embedded in many other bands, and thus evade direct detection in ARPES, and are not relevant for the transport data.

Thus, the search for material realisations of (quasi)-2D Dirac semi-metals [17] in bulk, three-dimensional crystals continues. Based on the specific case of SrMnSb$_2$ presented here, this research exemplifies the effectiveness of combining different experimental and theoretical techniques in order to make a call as to the standing of a particular material as a candidate magnetic topological semi-metal. On a more general level, these data underline the relevance of useful materials design guidelines (band fillings, crystal symmetries, absence of magnetism) for successfully finding protected 2D Dirac states in 3D, bulk crystals.

# Acknowledgements

We thank C. Kane and R.J. Slager for very valuable discussions at the outset of this project. We are grateful to Y. Pan and J. Beekman for preliminary analysis of transport and soft X-ray absorption data, respectively, and H. Luigjes and D. McCue for skilled technical support. This work is part of the Dutch national research programme 'Topological Insulators' of the Foundation for Fundamental Research on Matter (FOM), which is part of the Netherlands

Organization for Scientific Research (NWO). J.v.W. acknowledges support from the NWO Vidi program. We thank Diamond Light Source for access to beamlines I05[I09] (proposal numbers SI12969,SI15189[SI13743]) that contributed to the results presented here. Research carried out at BESSY-II was under proposal number 15203055ST, and was supported by the European Community's Seventh Framework Programme (FP7/2007-2013) under Grant Agreement No. 312284 (CALIPSO).

## A  Appendix I - tight binding model

Our tight binding model includes the bands arising from the plane of Sb1 atoms in SrMnSb$_2$, pictured in Fig 1(b) in the main text. Due to the orthorhombic distortion in this plane, there are two Sb atoms per unit cell. This plane can be modelled by two identical rectangular sublattices (*A* and *B*, shown using different shading in Fig. 1(b)), with distortions along two directions: an out-of-plane buckling along the crystallographic *a* axis that puts one sublattice slightly above the other; and an in-plane shift of the two sublattices with respect to one another along the *c* axis. The magnitudes of these shifts in the real material were determined by neutron diffraction on single crystals to be 0.2% of *a* and 3.2% of *c*, respectively [15].

We note that the lattice of the Sb1 plane is the same as one of the lattices that Young and Kane has modelled in Ref. [17]. However, where they considered a lattice with only *s* orbitals, each atom in our lattice has valence *p* orbitals. We choose the quantisation axes of these orbitals such that $p_x$ and $p_y$ lie precisely in between the crystallographic *b* and *c* axes. Two sublattices with each atom hosting three orbitals gives us a six-band model. Including spin, the 12-band, tight binding Hamiltonian for SrMnSb$_2$ can be expressed as follows:

$$\hat{H} = \hat{H}^{nn} + \hat{H}^{nnn} + \hat{H}^{so} + \hat{H}^{eso} + \hat{H}^{B}. \tag{3}$$

In order of greatest influence on the band structure, these terms represent the effects of: nearest neighbour hopping (*nn*); next-nearest neighbour hopping (*nnn*); atomic spin-orbit coupling (*so*); effective next-nearest neighbour spin-orbit interaction (*eso*); and magnetism (*B*). The first two terms are the hopping terms, which are spin-independent. Nearest neighbour hopping describe hopping between the two sublattices, described by

$$\hat{H}^{nn} = \hat{H}^{AB} + \hat{H}^{BA}, \tag{4}$$

with matrix elements

$$H^{AB}_{\mu,\nu} = \sum_{i=1}^{4} e^{i\mathbf{k}\cdot\mathbf{R}_i} t_{\mu\nu}(\mathbf{n}_i). \tag{5}$$

where $\mu$, $\nu$ are the atomic orbitals, $\mathbf{n}_i = \mathbf{R}_i/|\mathbf{R}_i|$ the unit vector connecting neighbouring atoms and $t_{\mu\nu}$ the hopping amplitudes determined using the Slater-Koster method, with the bond orientation taken into account [39]. Next-nearest neighbour hopping terms describe the hopping within a single sublattice:

$$\hat{H}^{nnn} = \hat{H}^{AA} + \hat{H}^{BB}, \tag{6}$$

where the matrix elements are given by

$$H^{AA}_{\mu,\nu}(k) = \left( \epsilon_\mu + \sum_{j=1}^{4} e^{i\mathbf{k}\cdot\mathbf{R}_j} t_{\mu\nu}(\mathbf{n}_j) \right) \delta_{\mu\nu}, \tag{7}$$

where $\epsilon_\mu$ is the chemical potential. Hopping between different orbital flavours is disallowed by symmetry. The Hamiltonian is diagonal in its spin indices.

The next component we include is the atomic spin-orbit coupling (SOC), with matrix elements

$$H_{\mu,\nu}^{so} = \lambda_{so} \langle \mathbf{L} \cdot \mathbf{s} \rangle_{\mu,\nu,\sigma,\sigma'},\tag{8}$$

where $\mathbf{L}$ is the angular momentum operator, $\mathbf{s} = (\sigma_x, \sigma_y, \sigma_z)$ is the vector of Pauli matrices representing the spin of the electrons, $\sigma$ and $\sigma'$ runs over different spins and $\lambda_{so}$ paramatrises the strength of the SOC. The resulting terms for a lattice with $p$ orbitals is given explicitly in Ref. [40]. Young and Kane's TB model, being for $s$ orbitals, does not include this (atomic) SOC [17].

However, Young and Kane *do* include an effective next-nearest neighbour SOC which is non-zero when there is no inversion symmetric point between the two next-nearest neighbouring sites, and leads to splitting of bands along X−M and Y−M [17]. In a semiclassical sense, a next-nearest neighbour hopping can be seen as traversing via the nearest neighbours, which entails taking a curved path. This generates an effective magnetic field which can then interact with the electron spin. For every allowed next-nearest neighbour hopping on each sublattice, we add a term to the Hamiltonian given by

$$H_{\mu,\nu}^{eso} = i \sum_{i=1}^{2} \sum_{j=1}^{4} e^{i\mathbf{k}\cdot\mathbf{R}_j} t_{\mu,\nu}^{eso} \left( \mathbf{R}_i \times \mathbf{R}_j \right) \cdot \mathbf{s},\tag{9}$$

where $\mathbf{R}_i$ are the two possible nearest neighbour connections that can be used as a first step on the way to next-nearest neighbour $\mathbf{R}_j$. The strength of the effective SOC is parametrised by $t_{\mu,\nu}^{eso}$.

Finally, we include canted anti-ferromagnetism observed in SrMnSb$_2$ as shown in Ref. [15]. In the Sb1 plane, it manifests itself as a sublattice-dependent field with moments aligned along $\pm\mathbf{a}$, the magnitude of which are given by a parameter $B_{AFM}$, together with a ferromagnetic field along $\mathbf{c}$ parameterised by $B_{FM}$:

$$\hat{H}^B = \hat{H}^{B_{AFM}} + \hat{H}^{B_{FM}}.\tag{10}$$

The matrix elements from these two fields are given by

$$\begin{aligned} H_{\mu,\nu}^{B_{AFM}} &= (-1)^i B_{AFM} \sigma_z \delta_{\mu\nu}, \\ H_{\mu,\nu}^{B_{FM}} &= -B_{FM} \sigma_x \delta_{\mu\nu}, \end{aligned}\tag{11}$$

with $i$ zero or one, depending on the sublattice. As was already mentioned in the main text, the band splitting due to magnetism is extremely small for realistic values of $B_{AFM}$ and $B_{FM}$, only becoming visible for moments on the order of several hundred $\mu_B$. Nonetheless, it breaks time-reversal symmetry and therefore leads to the loss of non-symmorphic symmetry protection as described in Ref. [17].

The TB parameters were at first adjusted to fit the DFT band structure and were later fine-tuned to fit to the ARPES data along the high symmetry directions shown in Fig. 5(b). Even though the relative simplicity of the TB model does not allow the fit be exact throughout the entire Brillouin zone, it does reproduce its main features very well, and captures the essence of the problem sufficiently to validate the Berry phase and gapping conclusions drawn from this approach. The Y-centred energy contour in the TB appears more extended along the $k_x = k_y$ direction than in the ARPES data for binding energies of 100 and 150 meV. Looking at the upper right quadrant of the (unsymmetrized) experimental data for an energy of 200 meV, the star-like shape is clearly evident also in the ARPES data.

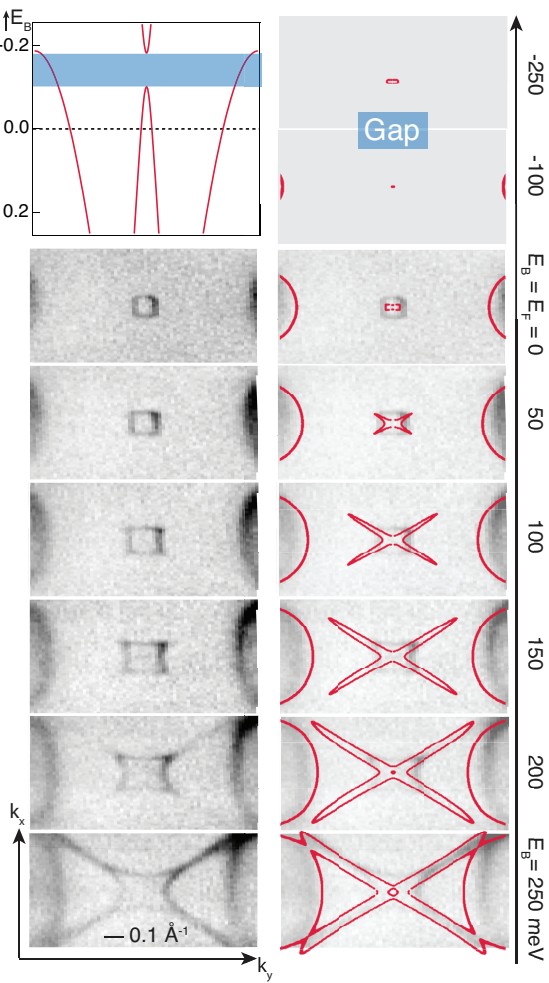

**Figure A1: Constant energy contours around the Y-point from ARPES and TB calculations**
Top-left panel: dispersion of the Y states along Γ—Y direction from the TB calculation; Bottom-left panel: constant energy ARPES maps around the Y-point at the binding energies indicated; Right panel: Constant energy contours of the bands from the TB calculations, integrating 5 meV above and below the energy shown, is superimposed on the top of ARPES data.

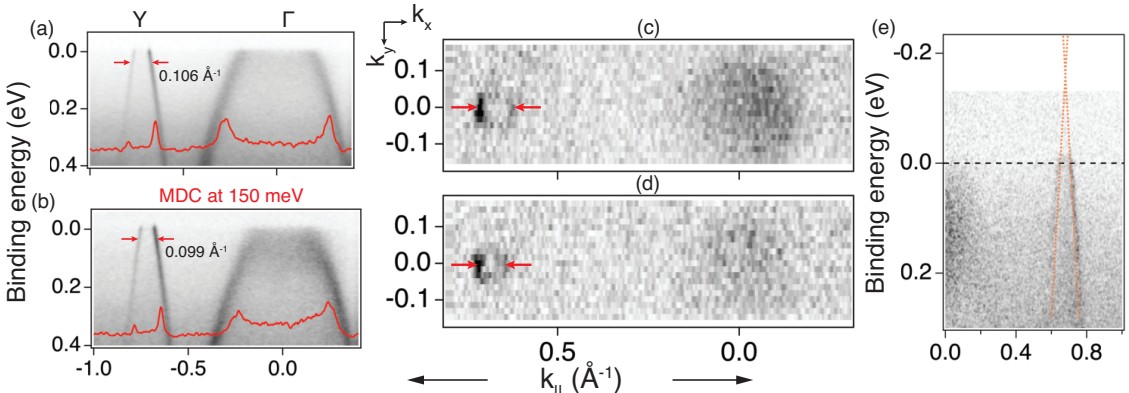

Figure A2: **Alkali metal decoration shifts Fermi level upward** ARPES data recorded along Γ- X at the I05 beamline before and after K metal was evaporated onto the cleavage surface of SrMnSb$_2$ at 10 K. Panels (a) and (b) show $I(E,k_{||})$ images, recorded along Γ— X, before and after the first K evaporation, respectively. The red traces are overlays of MDC's for E=150 meV. Panels (c) and (d) are from a different cleave, and show constant energy maps for E=50 meV, before and after another K evaporation experiment. For both K evaporation experiments [(a) vs. (b), or (c) vs. (d)], red arrows indicate the decrease in the $k$-space separation between the left and right branches consistent with an upward shift in E$_F$ on K decoration. Panel (e) shows an $I(E,k_{||})$ image for a Γ—X cut for the most shifted situation arrived at via K evaporation. The orange dashed lines extrapolate towards a putative crossing, now at ca 140 meV above E$_F$. All data are measured with 88 eV photon energy at 10 K.

## B Appendix II - chemical potential shift via *in-situ* K-adsorption

From the surface characterisation section in the main part of the paper, the Sb1 plane seems to be termination plane of our cleave. Given the relatively poor conductivity of SrMnSb$_2$ along the interplanar direction, we considered it worthwhile to try shifting the Fermi energy for the surface region of the crystal upwards (with respect to the band structure) by decoration with potassium atoms at low temperature. Panel (e) of Fig. A2 shows that E$_F$ can be shifted upward by maximally 60 meV in this manner (as the putative band crossing arrived at from the linear extrapolation of the occupied part of the Y-state bands demonstrates, in comparison to Fig. 4(a) in the main text). More K evaporation beyond the stage shown in Fig. A2(e) simply led to complete attenuation of the Sb1-related emission features. A 2D layer of ionised K atoms could either induce a surface n-type doping or be a source of downward bandbending. Thus the microscopic physics underlying the upward Fermi level shift we see on K decoration cannot easily be elucidated based on the data on hand. In either case, this Fermi level shift, due to charge transfer from the K atoms to the surface region of the SrMnSb$_2$, naturally leads to shrinkage of the $k$-space separation of opposite branches of the Y-states as can be seen comparing panels (a) with (b), and (c) with (d) in Fig. A2. Thus, this E$_F$ shift helps to reveal the previously unoccupied states, however, it is insufficient to uncover the gapping expected from the theory treatment of the Y-states discussed in the main text.

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
