# Peer review of "Electronic structure of the candidate 2D Dirac semimetal SrMnSb2: a combined experimental and theoretical study"

_SciPost Physics, doi:SciPost Phys. 4, 010 (2018)_

## Round 2 · Referee Report · Anonymous · 2017-12-27

Strengths

1- Provision of first ARPES data on SrMnSb2, a material which is currently in the focus of interest. The presented ARPES gives complementary information to existing, mostly magnetotransport literature and allows the authors to challenge recent claims that SrMnSb2 is a representative of a magnetic topological semimetal (J. Y. Liu et al., Nature Materials 2017).
2- The authors present high-field magnetotransport data on one and the same SrMnSb2 crystals studied also by ARPES. It allows a direct quantitative comparison of quantumtransport with the observed Fermi surface structure in ARPES data.
3 - By comparison of two theory approaches (Tight binding and ab intio DFT) the authors shed light on a possible correlation of topology and a time reversal symmetry breaking due to a canted antiferromagnetic state as suggested by J. Y. Liu et al., Nature Materials 2017.
iu et al., Nature Materials 2017.

Weaknesses

1 - The referee believes that the author`s conclusion that bulk transport is governed by Y-states only leaves some questions unanswered.
2 - The discussion of the kz-dispersive behavior of states at the Gamma point remain undecisive regarding their dimensionality (2D vs bulk).

Report

The manuscript shows experimental data and calculations of high significance on a very interesting topic. It is generally well written and evaluation/interpretation steps are easy to follow. Figures are well done.
Publication is highly recommended provided the following comments/questions are addressed:

1 - Possible band bending effects on the filling of 2D states vs bulk states: In the main text the authors comment that "no significant time effects" were observed on the time scale of days. Nevertheless, band bending effects can be instantly present after cleaving. In a second step, K evaporation can alter such band bending effects, and thus observed shifts in the Fermi level of Y states might not necessarily only due to the donor behavior of the alkali metal.

2 - The authors describe a cleaving plane within the Sr - Sb1 - Sr unit as "most likely", and comment that atomic distances in STM "match well" a Sb1 terminating plane. If the referee understands well, Sr termination is just as likely. Are there also areas visible in STM where the surface is Sr terminated? What influence would alternating cleaving areas have on the average ARPES, e.g. by local variations in band bending?

3 - The authors claim that the Fermi surface visible in ARPES data transport perfectly matches the frequency of SdH oscillations and "unambiguously link the Y-states to the conductivity". The referee agrees with the former statement but points out that it is not surprising that broadened states around the Gamma point most likely will not appear well defined in the FFT of the SdH data. It is thus not clear how much the states at Gamma will contribute . Their mobility may be low as the authors speculate but DOS at EF high.
Does the the hole carrier density and average slope in Fig.2(f) fit to transport through exclusively the Fermi surface given by Y-states?

4 - From the kz dispersion in the Y-T direction (Fig. 5) the authors follow that Y-states are of 2D character. At the same time - also from Fig. 5 - a 2D character of states at the Gamma point are claimed from cuts along T-Z and Gamma-Y directions. The latter argument is not clear to the referee maybe due to a lack of knowledge. Along the Gamma-Z direction a dispersion is clearly visible, which would speak against a 2D character?

5- Ref.[15] stresses the fact that dHvA data is more reliable than SdH frequencies regarding the Fermi surface for 2D-like materials. For SrMnSb2 deviations between SdH and dHvA by 20% was observed. The authors could comment in the manuscript on the respective error margins in the parameter A in the Onsager Eq. when evaluating their SdH data.

6 - In the discussion of the ARPES data the authors comment on the presence of 90deg domains visible in ARPES. The expected influence of such domains on non-local average transport properties should be commented on, in particular on the relative orientation of canted AFM phases between domains. What is the length scale of such domains with respect to a phase coherence and Berry phase length scales?

Requested changes

1 - A comment/discussion of possible band bending effects on the filling of 2D states vs bulk states is necessary, referring to points 1 and 2 in the report above.

2 - The referee believes that in order to claim that transport is due to Y-states only, further evidence or explanation should be provided (see point 3 in the report above). Does the derived hole carrier density from the average Hall slope in Fig.2(f) fit to a DOS estimation solely from Y-states at the Fermi level?
If not, how much would possible transport contributions from Gamma states affect the Berry phase estimation from LLs?

3 - A possible 3D character of states at the Gamma point should be further addressed (see point 4 in the report above). Up to now a possible kz-dispersion of states at the Gamma point in Fig. 3(a) could not be excluded according to the author`s discussion. From the presented theory, the statement that Gamma states are of 2D type from Fig. 5 (e.g. along the Gamma-Z direction) is - at least to the referee`s perception - not obvious.

4 - The authors describe a cleaving plane within the Sr - Sb1 - Sr unit as "most likely", and comment that atomic distances in STM "match well" a Sb1 terminating plane. It has to be commented if Sr termination after cleaving is equally likely or even observed e.g. in STM (see point 2 in the report above).

5 - A discussion or short comment on the influence of domains in non-local average transport properties should be included (see point 6 in the report above). What is the relative orientation of canted AFM phases between such domains? What is their length scale with respect to phase coherence and Berry phase length scales?

  • validity: good
  • significance: high
  • originality: good
  • clarity: high
  • formatting: excellent
  • grammar: excellent

Author:  Shyama Varier Ramankutty  on 2018-01-24  [id 205]

(in reply to Report 1 on 2017-12-27)
Category:
answer to question
correction

Our thanks to the referee for her/his positive assessment of the paper, and recommendation to publish. The referee reports and our answers to each of them are given below.

1 - Possible band bending effects on the filling of 2D states vs bulk states: In the main text the authors comment that "no significant time effects" were observed on the time scale of days. Nevertheless, band bending effects can be instantly present after cleaving. In a second step, K evaporation can alter such band bending effects, and thus observed shifts in the Fermi level of Y states might not necessarily only due to the donor behaviour of the alkali metal. Our answer to the first part of the question: From research on other Dirac systems and related materials, we have direct experience of how a difference of the Fermi level position in the (near) surface region and in the bulk due to band bending like phenomena makes itself felt in ARPES. Example 1: in bulk-insulating Bi-based 3D topological insulators such as Bi${2-x}$Sb$_x$Te$$Se$_y$ (x=0.58, y=1.3) the 3D carrier density from Hall data is ~1x10$^{16}$cm$^{-3}$ (see for example the table on p7 of Y. Pan et al., New Journal of Physics 16 123035 [2014]), and there diverse downward and upward BB effects has been seen in ARPES (see Frantzeskakis et al., PRB 91, 205134 [2015] and Franzeskakis et al., PRX 7, 041041 [2017]). The timescales of the downward BB and the upward surface photovoltage and/or photon-stimulated desorption shifts are such that they are very clearly visible in the data when doing ARPES in the same way as we have done here on SrMnSb$_2$ in the SciPost submission. Example 2: we also saw clear time-dependent changes in the position of the Fermi level in ARPES of YbB$_6$ (Franzeskakis et al, PRB 90, 235116 [2014]), and there is a consensus that these are band bending. Thus, the fact that the ARPES data were stable over long periods of time is – in our experience of measuring ARPES of >100 solid systems – already a good indication that BB effects are not playing a major role. Supporting this is the fact that – for SrMnSb$_2$ - the k$_F$ values and energy positions of the features at both $\Gamma$ and Y are fully reproducible from the cleave to cleave and when measured at different beamlines (BESSY vs. Diamond Light Source). This would generally not be the case if BB were a serious issue. Finally, we provide two transport-related arguments. Firstly, the 3D carrier density in SrMnSb$_2$ is of order 10$^{19}$ cm$^{-3}$, three orders greater than in the 3D TI BSTS which shows band bending, thus SrMnSb$_2$ is simply too conducting/metallic for band bending. Secondly, ARPES sees the top nm of the crystal and transport is a bulk probe. Thus, the fact that the ARPES of the Y states agrees so well with the QO data from the transport (either that from Liu et al. or our own) would be very difficult to fit in a picture in which BB plays a role in the ARPES data. Our answer to second part of the question: from the above arguments, we are happy that there is no significant band bending for the pristine cleavage surface of SrMnSb$_2$. Thus, adsorbing K cannot alter a pre-existing band bent situation. The microscopic physics underlying the upward Fermi level shift we see on K decoration of SrMnSb$_2$ cannot easily be elucidated based on the data on hand. On a pragmatic level, the answer to this question does not really matter, as the main profit from the ARPES point of view is that the K-decoration enables us to see ‘further up’ into the formerly unoccupied states. As argued above, the relatively large 3D carrier concentration in SrMnSb$_2$ (compared, for example, to a good 3D topological insulator) would suggest K-induced band bending may be unlikely. However, we accept that a 2D layer of ionised K atoms could be a source of downward band bending, were the overall carrier density low enough not to screen those charges. \textbf{Changes made:} 1. We modified the text at the end of Section I (where the term band-bending cropped up), adding a suitably condensed form of the main points we argue above. – on p5 2. In Appendix II (K-decoration) we modify the text slightly to recognise that (a) the mechanism of the Fermi level change is not the main issue (the uncovering of previously unoccupied states is) (b) a surface n-type doping and K-induced band bending and two ways of looking at how the E$_F$ shift could come about.

2 - The authors describe a cleaving plane within the Sr - Sb1 - Sr unit as "most likely", and comment that atomic distances in STM "match well" a Sb1 terminating plane. If the referee understands well, Sr termination is just as likely. Are there also areas visible in STM where the surface is Sr terminated? What influence would alternating cleaving areas have on the average ARPES, e.g. by local variations in band bending? Our answer: The STM data we have do not show atomic resolution with distances matching a Sr termination. We agree that if the STM side of the cleave is Sb1 terminated then the cleavage post side should involve Sr termination. More detailed STM studies would be needed for this and they go beyond the scope of this paper. Changes made: we add a sentence that from symmetry considerations also Sr termination is possible, but that we do not have any concrete experimental evidence for this. – on p5

3 - The authors claim that the Fermi surface visible in ARPES data transport perfectly matches the frequency of SdH oscillations and "unambiguously link the Y-states to the conductivity". The referee agrees with the former statement but points out that it is not surprising that broadened states around the Gamma point most likely will not appear well defined in the FFT of the SdH data. It is thus not clear how much the states at Gamma will contribute. Their mobility may be low as the authors speculate but DOS at EF high. Does the hole carrier density and average slope in Fig.2(f) fit to transport through exclusively the Fermi surface given by Y-states?  Our answer: The referee agrees with us that the Y-states seen in ARPES are – beyond doubt – linked to the SdH data. The latter type of data (which strongly resemble those presented here in terms of the observed frequencies) are used in Ref. 15 to argue for topologically non-trivial behaviour in SrMnSb$_2$. Whether the states at Gamma contribute to the resistivity or Hall carrier density is a separate (and less important) issue than that of the topologically trivial or non-trivial nature of the states at Y, as it is these latter states (and not those at Gamma) that drive the SdH oscillations. Changes made: We modify the text to connect the ARPES states seen at the Y-point with the QO experimental data from which Berry phase analyses of this materials are possible: “….and importantly, this enables us to unambiguously link the Y-states to the quantum oscillation data in the magnetotransport in high-quality single crystals of this material”. – on p8

4 - From the kz dispersion in the Y-T direction (Fig. 5) the authors follow that Y-states are of 2D character. At the same time - also from Fig. 5 - a 2D character of states at the Gamma point are claimed from cuts along T-Z and Gamma-Y directions. The latter argument is not clear to the referee maybe due to a lack of knowledge. Along the Gamma-Z direction a dispersion is clearly visible, which would speak against a 2D character? Our answer: Apologies that our original text was insufficiently clearly formulated, and thanks to the referee for making this clear to us. Changes made: The text now reads: - on p12 The DFT data show the states close to E$F$ to be independent of k$_z$ between the Y and T points (see the inset to Fig. 2a for the Brillouin zone labels), indicating the 2D nature of the Y-states as shown in the ARPES data of Fig. 3. The DFT data of Fig. 5a shows that two-thirds along both the Y-$\Gamma$ (k$_z$=0) and T-Z (k$_z$=pi) k$$-directions there is no significant lifting of the degeneracy of the two bands involved. If this also applies to k$z$ values between these extrema, then this can explain the ‘straight-up’ edges seen in the k$_z$ dependent ARPES data at plus/min 0.2 \AA$^{-1}$ either side of $\Gamma$ in Fig. 3a. However, for k$$ =0 the DFT data of Fig. 5a do show dispersion along k$_z$ (between $\Gamma$ and Z), with the two bands being degenerate at Z and not so at $\Gamma$.

5- Ref. [15] stresses the fact that dHvA data is more reliable than SdH frequencies regarding the Fermi surface for 2D-like materials. For SrMnSb2 deviations between SdH and dHvA by 20\% was observed. The authors could comment in the manuscript on the respective error margins in the parameter A in the Onsager Eq. when evaluating their SdH data. Our answer: In Ref. 15 (bottom of p2) the SdH frequency is given as 67 T for the correct field direction, and this agrees with the frequency for the analogous Fermi surface from dHvA data (top of p3 of Ref. 15). The authors of Ref. 15 state that the SdH data fit the expectations of LK theory less well than the dHvA data do. The error bar in the determination of the Fermi surface area from Onsager relation is of the order of 5\%, as obtained from the standard deviation of the Fourier transform peaks of the SdH data. Changes made: Modified the text to add the error bar in the estimation of Fermi surface area from Onsager relation (see p.8 of the modified manuscript).

6 - In the discussion of the ARPES data the authors comment on the presence of 90deg domains visible in ARPES. The expected influence of such domains on non-local average transport properties should be commented on, in particular on the relative orientation of canted AFM phases between domains. What is the length scale of such domains with respect to a phase coherence and Berry phase length scales? Our answer: The ARPES data do not image domains, so we do not have any detailed data on domain sizes to answer this question. The fact that the data from BESSY did catch a single domain would suggest a domain can be of 10 $\mu$m or more in size: they are macroscopic objects. Within each of these domains, the effective moment from the cAFM order points along b, and the Berry phase analysis done using the TB model for the states around Y will be equally valid inside each domain. We are not aware of any transport data out there measuring transport along the b-direction and comparing it to the c direction (i.e. the two in plane directions), nor of any examples of how possible scattering at the (few) boundaries there are between such large (10’s of micron) objects leads to de-phasing impacting quantum transport experiments. Changes made: We add a brief footnote that we do not expect the domain boundaries across which the current is flowing to influence the quantum transport phenomena being analysed in the QO experiments (see p.7/p.8 of the modified manuscript).

Requested changes: We have given an account of the changes made in each of the responses above.

---

## Round 2 · Referee Report · Anonymous · 2017-12-31

Strengths

comprehensive experimental and theoretical work

Weaknesses

manuscript is extremly long - could have been written more concisely

Report

The authors present a combinded experimental and theoretical study on SrMnSb2. The data are interesting and the conclusions are solid. I recommend publication of this manuscript once the requested changes have been taken into account.

Requested changes

1) Emphasis on the major findings is necessary in the abstract
2) theoretical procedure - how are these assumptions justified: spin-orbit is included as a perturbation and the value of the magnetic moment?
3) Is the crystallograhic unit cell the chemical unit cell here?
4) page 10 (intermediate conlusion): add the arguments here. I suppose the authors do means that based on their observations so far, the systems seems to be gapless.
5) is there a beating pattern in the transport data, i.e. a second band? - please comment

  • validity: high
  • significance: good
  • originality: high
  • clarity: high
  • formatting: excellent
  • grammar: excellent

Author:  Shyama Varier Ramankutty  on 2018-01-24  [id 206]

(in reply to Report 2 on 2017-12-31)
Category:
answer to question

Our thanks to the referee for her/his positive assessment of the paper, and recommendation to publish.

Requested changes: 1) Emphasis on the major findings is necessary in the abstract Our answer: we worked really hard to get the abstract this concise, and after the brief text giving the context of SrMnSb$_2$ as a material – which we feel is important for potential readers to identify this paper as suitable for them to read - everything in the abstract is ‘major findings’. For example, the XAS data are not described in the abstract, so we did not include everything.

2) theoretical procedure - how are these assumptions justified: spin-orbit is included as a perturbation and the value of the magnetic moment? Our answer: Spin-orbit coupling is treated as a perturbation term with the scalar-relativistic orbitals as basis. This is a well-established way to implement spin-orbit coupling effect in most DFT codes such as Wien2k and VASP (see A. H. MacDonald et al. J. Phys. C: Solid State Phys. 13 2675 [1980]). The value of the magnetic moment about 5 $\mu$B/Mn atom is obtained by our self-consistent DFT calculations. Changes made: To avoid possible confusion, we modify the related sentence adding the points mentioned here and the related reference (top of p4 of the modified manuscript).

3) Is the crystallograhic unit cell the chemical unit cell here? Our answer: the crystallographic unit cell contains four chemical formula units.

4) page 10 (intermediate conclusion): add the arguments here. I suppose the authors do means that based on their observations so far, the systems seems to be gapless. Our answer: thanks to the referee for pointing out the text can be clearer. Changes made: the word ‘gapless’ has been added (p.10 of the modified manuscript).

5) is there a beating pattern in the transport data, i.e. a second band? - please comment Our answer: There is no beating in the $\rho_{xx}$ and $\rho_{xy}$ for the measured field range, neither do Liu et al. (Ref. [15]) report this. However, combining these into $\sigma_{xx}$ shows a beating. This a not due to two different fundamental frequencies, possibly it can be attributed to an inhomogeneity in the carrier concentration.

---

## Round 2 · Referee Report · Anonymous · 2018-1-8

Strengths

1 - Previous reports, claiming topological states in SrMnSb2, are quite indirect [ref 15 in the manuscript]. The present manuscript present the first direct k- and energy- resolved band structure as extracted by ARPES.

2 - The authors combine theory and various experiments to prove that the states near to the EF in SrMnSb2 are topologically trivial, in disagreement with the previous study [15].

3 - The manuscript is well written, detailed and the results are of broad interest.

Weaknesses

1 - Tight binding band TB structure deviations from experimental ARPES band dispersion are not negligible. So the claims on the Berry phase from TB being zero maybe not fully conclusive. However, the Landau level diagramme in Fig. 7 (b) shows that the experimental Berry phase is zero, in line with the same result from TB.

Overall I find the claim of zero berry phase convincing even if the TB model for the band structure is not perfect.

Report

This manuscript presents a combined experimental and theoretical investigation of the electronic structure of
SrMnSb2. The main conclusion of the manuscript is that electronic states near the Y-point are gapped above EF, implying trivial topology. This result is in contrast with previous results which suggested that SrMnSb2 is a magnetic topological semimetal.

In general, I believe that the manuscript is well presented and the results are of broad interest.In particular, the authors present a very detailed and comprehensive study of SrMnSb2, with both theory and experiment supporting their claim of trivial states in this system.

For these reasons I believe that the manuscript meets the SciPost cryteria of impact and I suggest the publication in SciPost.

Requested changes

1 - In the abstract the authors state that the agreement with DFT and TB is excellent. While I agree that the agreement with DFT is excellent (Fig 6), I cannot say the same for the TB (Fig A1). Similarly, at the end of Appendix A the authors state that "Fig. A1 shows good agreement between the ARPES and TB constant energy contours."

I suggest that the authors soften these claims (something like "TB and ARPES are in qualitative agreement" should work).

  • validity: high
  • significance: high
  • originality: good
  • clarity: good
  • formatting: excellent
  • grammar: excellent

Author:  Shyama Varier Ramankutty  on 2018-01-24  [id 208]

(in reply to Report 3 on 2018-01-08)
Category:
answer to question

Our thanks to the referee for her/his positive assessment of the paper, and recommendation to publish.

1 - In the abstract the authors state that the agreement with DFT and TB is excellent. While I agree that the agreement with DFT is excellent (Fig 6), I cannot say the same for the TB (Fig A1). Similarly, at the end of Appendix A the authors state that "Fig. A1 shows good agreement between the ARPES and TB constant energy contours."
I suggest that the authors soften these claims (something like "TB and ARPES are in qualitative agreement" should work).

Our answer: Given the (advantageous) simplicity of the TB approach the agreement with the experimental reality is satisfying, but we are happy to soften the text in both the abstract and appendix.
Changes made: we modify the text in the abstract from ‘excellent’ to ‘good’ agreement. In the appendix, we now mention (on p. 20 of the new manuscript) (i) the DFT agreement with the experimental data is good and (ii) while the TB model shows larger deviations from the experimental data, but still captures the essence of the problem sufficiently to validate the Berry phase and gapping conclusions drawn from the TB approach.

---

## Editorial Decision

published